# ControlMLLM: Training-Free Visual Prompt Learning for Multimodal Large Language Models

**Mingrui Wu**[1], **Xinyue Cai**[1], **Jiayi Ji**[1]*, **Jiale Li**[1], **Oucheng Huang**[1],
**Gen Luo**[1], **Hao Fei**[2], **Guannan Jiang**[3], **Xiaoshuai Sun**[1], **Rongrong Ji**[1]
1 Key Laboratory of Multimedia Trusted Perception and Efficient Computing,
Ministry of Education of China, Xiamen University, 361005, P.R. China
2 National University of Singapore    3 CATL
mingrui0001@gmail.com

## Abstract

In this work, we propose a training-free method to inject visual prompts into Multimodal Large Language Models (MLLMs) through learnable latent variable optimization. We observe that attention, as the core module of MLLMs, connects text prompt tokens and visual tokens, ultimately determining the final results. Our approach involves adjusting visual tokens from the MLP output during inference, controlling the attention response to ensure text prompt tokens attend to visual tokens in referring regions. We optimize a learnable latent variable based on an energy function, enhancing the strength of referring regions in the attention map. This enables detailed region description and reasoning without the need for substantial training costs or model retraining. Our method offers a promising direction for integrating referring abilities into MLLMs, and supports referring with box, mask, scribble and point. The results demonstrate that our method exhibits **out-of-domain generalization** and **interpretability**. Code: `https://github.com/mrwu-mac/ControlMLLM`.

## 1  Introduction

In recent times, there has been a surge in the development and adoption of large language models (LLMs), such as GPT-4 [1] and Llama [53], showcasing remarkable capabilities in addressing a wide range of human-generated questions. The success of these models has sparked interest among researchers in exploring the integration of LLMs with visual inputs. Consequently, a new class of models known as Multimodal Large Language Models (MLLMs) has emerged [36, 33, 17, 67, 81, 38]. However, despite their widespread adoption, traditional MLLMs often face limitations due to their reliance on coarse image-level alignments. This restricts users to guiding MLLMs solely through text prompts for detailed region description and reasoning. However, text often fails to capture the intricate visual nuances present in an image.

Addressing this challenge, recent efforts [68, 5, 75, 12, 37] have pioneered the integration of referring abilities within MLLMs, which enables users to provide input by pointing to specific coordinates of the objects or regions, as shown in Figure 1 (left). However, these endeavors typically entail substantial training costs to equip MLLMs with referring capabilities. Additionally, the model must undergo retraining to adapt to new data domains or new base MLLMs.

In this work, we propose a training-free method to inject the visual prompts into the Multimodal Large Language Models via learnable latent variable optimization. The method originates from our observation of the attention maps from the MLLM decoder, which model the relationship between

---

*Corresponding Author

38th Conference on Neural Information Processing Systems (NeurIPS 2024).

**Figure 1:** Comparison between the training method and our training-free method. The training method typically requires a large amount of in-domain data for training and cannot generalize to out-of-domain prompts. In contrast, our method can easily adapt to prompts from a new domain in a training-free manner.

the pixels and text prompt tokens and encompass rich semantic relations that significantly influence the generated text. However, MLLMs typically involve fine-tuning an MLP layer to bridge the gap between visual and linguistic representations, which means that the output of the MLP layer can indirectly impact the relationship between text prompt tokens and pixels in the attention layers of the MLLM decoder, thereby altering the model's output.

Thus, our key idea is that we can alter the outputs of MLLMs by adjusting the visual tokens from the MLP output during the inference process, controlling which text prompt tokens attend to which visual tokens in the attention layers. Specifically, we augment visual tokens with an additional learnable latent variable. Subsequently, we optimize the learnable latent variable based on an energy function designed to enhance the strength of the referring regions in the attention map between the text tokens and the visual tokens.

Our method enables referring MLLMs with various visual prompts, including box, mask, scribble and point, and does not require model training, fine-tuning, or extra data. We also demonstrate that our method exhibits out-of-domain generalization and interpretability.

## 2  Related Work

**MLLMs**  Motivated by the accomplishments of Large Language Models (LLMs) [1, 53], there is a burgeoning trend among researchers to develop a diverse range of Multimodal Large Language Models (MLLMs) [33, 36, 17, 67, 32, 38, 39, 18, 22, 78, 15, 58, 35, 20, 21, 19]. These MLLMs typically comprise a visual encoder, a language decoder, and an image-text alignment module. The visual encoder and the language decoder are often sourced from pre-trained models, such as CLIP [44], DINOv2 [41], Llama [53], and Vicuna [16]. Meanwhile, the image-text alignment module is trained on image-text pairs and fine-tuned through visual instruction tuning to enhance its visual conversation capabilities. These Multimodal Large Language Models (MLLMs) often confront limitations stemming from their reliance on coarse image-level alignments.

**Referring MLLMs**  In recent research, there has been a noticeable trend towards integrating foundation models with tasks involving referring dialogue. These models [69, 74, 75, 12, 37, 43, 68, 64, 71, 5, 73, 40, 26, 34, 70, 79, 11, 51, 46, 80, 63, 8, 24, 45, 52, 72] introduce spatial visual prompts as extra input and are trained using region-text pairs. By leveraging this approach, they effectively bridge the gap between textual prompts and visual context, enabling comprehensive understanding of

image content at the regional level. However, these methods inevitably require a substantial training burden.

**Training-free Control in Text-to-Image**    There are numerous works on controllable text-to-image generation, among which training-free methods [27, 14, 62, 30] are most relevant to our research. Among them, Prompt-To-Prompt [27] explore the role of attention in text-visual interactions in Stable Diffusion [47] model, while Layout-Guidance [14] indirectly bias attention in Stable Diffusion model by optimizing an energy function. These contributions significantly inform our investigation into enhancing controllability and interpretability in MLLMs.

**Visual Prompt**    The visual prompt can be categorized into two main techniques: hard prompt and soft prompt. The hard visual prompt works [48, 57, 66, 65] that direct the model's attention to the region or enable visual grounding abilities in the Multimodal Models in a training-free and convenient manner by directly manipulating images, such as color guidance [60, 23]. However, these methods inevitably compromise the structural information of the images, or a strong understanding of the corresponding patterns by the model is required. In contrast, the soft visual prompt works [28, 4, 77] integrate learnable visual prompts into models to adapt them for different downstream tasks. However, these methods do not support region guidance and require fine-tuning the model with downstream data. In contrast, we optimize a learnable latent variable to support referring MLLM in the test time, without any downstream training data required, and TPT [49] is most related to our work.

## 3    Background

**Multimodal Large Language Models (MLLMs):**    The MLLMs typically consist of a visual encoder, an LLM decoder, and an image-text alignment module. Given an image $I$, the frozen vision encoder and a subsequent learnable MLP are used to encode $I$ into a set of visual tokens $e_v$. These visual tokens $e_v$ are then concatenated with text tokens $e_t$ encoded from text prompt $p_t$, forming the input for the frozen LLM. The LLM decodes the output tokens $y$ sequentially, which can be formulated as:

$$y_i = f(I, p_t, y_0, y_1, \cdots, y_{i-1}). \tag{1}$$

Considering LLaVA-liked [36] MLLMs, the LLM in MLLMs typically employs a transformer model [54] with the attention layer as its core. In such model, the attention maps represent the relationships between the visual tokens and the text prompt tokens. The attention map in attention layer $\tau$, computed on the transformed visual-text concatenated embeddings $[e_v, e_t]^{(\tau)}$, is obtained as follows:

$$A^{(\tau)} = \text{softmax}(\frac{[e_v, e_t]^{(\tau)} \cdot ([e_v, e_t]^{(\tau)})^T}{\sqrt{d_k}}), \tag{2}$$

where $d_k$ is a scaling factor. $A^{(\tau)}$ consists of $A_{ij}^{(\tau)}$ with $i, j \in \{1, \cdots, n\}$, representing the relationship between the $i$-th token and the $j$-th token, and their impact on the output.

**Training Referring MLLMs:**    The objective of training referring MLLMs is to inject the visual prompt $r$ into the MLLMs to achieve referring ability via model parameter learning. The visual prompt $r$ can take various forms, such as a box, mask, scribble, or point, to indicate specific locations or regions within the image.

Current referring MLLMs typically need to be fine-tuned on a training set with positional annotations before they can be effectively used. The fine-tuning process involves maximizing the log likelihood of generating the text conditioned on $I$, $p_t$, and $r$ over the entire training dataset. This can be formulated as:

$$\theta^* = \arg\max_{\theta} \sum_{i=1}^{U} \log P(y_i \mid I_i, p_t, r, y_0, y_1, \cdots, y_{i-1}; \theta), \tag{3}$$

where $\theta$ represents the parameters of the model $f$, and $U$ is the number of samples in the training set. This method significantly enhances the model's fine-grained understanding and interactivity. However, it incurs high training costs. Additionally, such fine-tuning strategies result in domain-specific behaviors, which have been shown to compromise the out-of-distribution generalization and robustness of MLLMs [49]. Therefore, when domain shifts occur, the model needs to be retrained, leading to a lack of flexibility.

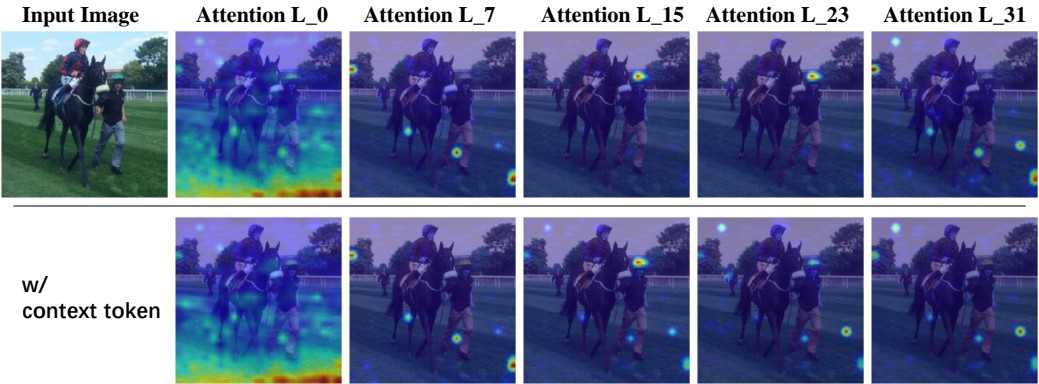

| Input Image | Attention L_0 | Attention L_7 | Attention L_15 | Attention L_23 | Attention L_31 |

w/ context token

Prompt: "What color is the **hat** the person is wearing?"
Output: "The hat the person is wearing is **green**."

Figure 2: The attention maps in various layers of the MLLMs, with the numbers indicating the respective layer indices. The top line visualizes the attention between the prompt token "hat" and the visual tokens, while the bottom line visualizes the attention between the context token (mentioned in Sec. 4.2) and the visual tokens.

## 4 Method

We aim to propose a training-free method to overcome the inconveniences of traditional training. Training-free referring MLLM maintains the model parameters $\theta$ frozen, eliminating the need for any training or fine-tuning with samples from the training set. During inference, the only information available is the single test sample without label information, as shown in Figure 1 (right).

In this section, we explore and design a solution to address the challenges of Training-free Referring MLLMs. The key task is to flexibly embed visual prompts during the inference phase while maintaining the model's reasoning capabilities. To begin with, we delve into the mechanism of MLLMs (see Sec. 4.1), our key observation is the attention mechanism in LLM capturing the relationship between the model's output and the input pixels. Further, the visual tokens inputted into the LLM influence the values of the attention maps to indirectly control the model output. Building on this analysis, we propose the Latent Variable learning (a test-time prompt tuning strategy [49], see Sec. 4.2) to edit the visual tokens, as shown in Figure 4. This method effectively integrates visual prompts into pre-trained MLLMs, enabling fine-grained visual reasoning.

### 4.1 Analysis of the Attention in LVLMs

We begin by analyzing *which factors in the model truly capture the relationship between input and output?* In other words, we seek to understand *how to interpret the association between the model's output and the input pixels*.

As demonstrated by Equation 1, Multimodal Large Language Models (MLLMs) fundamentally model the maximum likelihood output based on visual input and text prompts. By conditioning on the text prompt, the model can determine which parts of the image have the greatest impact on the output. Building on the discussions in the Sec. 3 and illustrations in the Figure 2 (top line), we can observe that the attention map models the influence of visual tokens on the output conditioned by the text prompt. Therefore, the attention map in MLLMs not only provides interpretability regarding the relationship between model output and input pixels but also facilitates guiding the model's output.

A natural idea is that we can directly alter the model's output by editing the attention maps. Inspired by IBD [82], we achieve this by adding an adjustment coefficient $\eta$ to the attention related to the visual tokens corresponding to the referring region, which can be formulated as,

$$A^{(\tau)} = \text{softmax}\left(\frac{[e_v, e_t]^{(\tau)} \cdot ([e_v, e_t]^{(\tau)})^T}{\sqrt{d_k}} + M\right),$$

$$M_i = \eta \quad \text{if } i \text{ in } r \quad \text{else } 0,$$

(4)

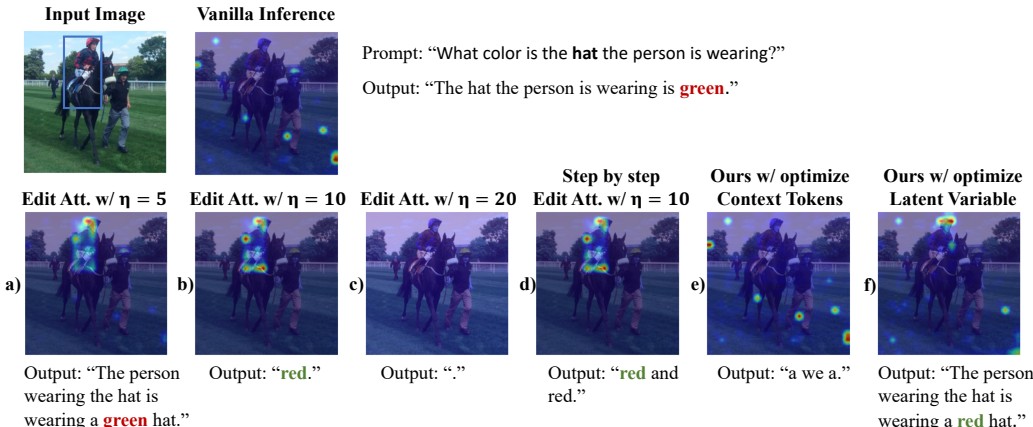

Figure 3: Manipulating attention with various methods: (a), (b), and (c) demonstrate the manipulation of the attention map by adding an adjustment coefficient $\eta$ on the attention map in the first step during the model inference. (d) illustrates the step-by-step editing approach. (e) showcases that optimizing a learnable context tokens (mentioned in Sec. 4.2) instead of visual tokens, while (f) presents the results of our method optimizing the learnable latent variable.

where $M$ is a mask with the same shape as the attention map, $r$ denotes the referring region. However, we have to carefully select a suitable coefficient $\eta$ for each example. When the $\eta$ is too small, it leads to ineffective control (as shown in Figure 3 a), and when it is too large, it can impact the language capabilities of the LLM (as shown in Figure 3 c). Additionally, we found that it is sufficient to manipulate the attention map at the 0-th step during model inference (as shown in Figure 3 a,b,c), as it is most directly associated with the text prompt, and manipulating attentions step by step also affects the expression of the LLM (as shown in Figure 3 d). Overall, directly manipulating attention maps is not a viable approach because it overlooks the relationships between attention layers and not all layers' visual tokens decide the output [13].

We note that in the most MLLMs, typically the MLP layer is trained for image-text alignment. This implies that MLLMs indirectly affect the values of the attention map by learning the parameters of the MLP layer to alter the visual tokens. In other words, the visual tokens inputted into the LLM directly influence the values of the attention maps.

It is also worth noting that the input text prompt also directly influences the model's output, particularly regarding non-visual-related [76] output content. However, we aim to explain the correlation between the output and the input image. Therefore, we do not consider the direct impact of the text prompt on the output in our analysis.

## 4.2 Manipulating Attention via Latent Variable Learning

Based on the analysis above, our core idea is to indirectly influence the attention maps by editing visual tokens, thereby focusing on the referred regions. We achieve this by optimizing a learnable latent variable based on an energy function [14, 59], which calculates the relationship between the input referring and the attention maps. To do this, we first need to determine which attention maps to use. One approach is to use attention maps between each text prompt token and all visual tokens. However, because visual tokens typically have a significant impact on the result based on only a few most relevant text prompts (referred to as **highlight text tokens**), using all attention maps would be computationally redundant. Yet, for users, identifying the highlight text tokens can be challenging. Therefore, we simply average pool the attention maps generated for each text prompt token to represent the global context of the text prompt (referred to as the **context token**) and its association with visual tokens. We found that this simple method of using context tokens produces attention maps similar to those generated by highlight text tokens, as shown in Figure 2 (bottom line). We leave the optimization based on highlight text tokens for future work.

Specifically, our method supports four types of referring shapes, including box, mask, scribble, and point. We employ two types of energy functions to respectively support these referring shapes: a hard

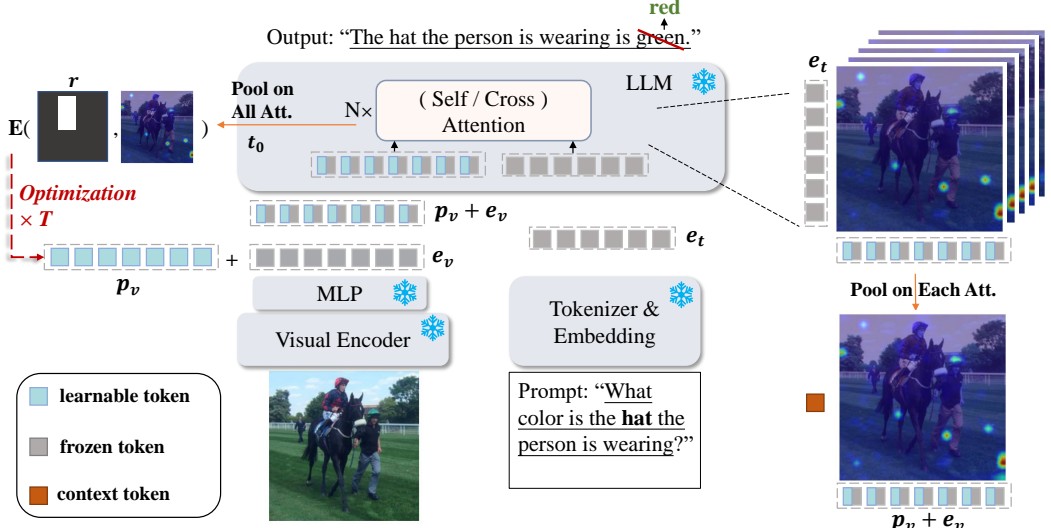

Figure 4: The overview of our method. With the provided visual prompt, we convert it into a mask, and compute the mask-based energy function between the mask and the pooled attention map. During the inference process, we conduct backpropagation to optimize a learnable latent variable. This process is executed at the 0-th step of model inference and iterated $T$ times.

mask-based energy function for box and mask referring, and a soft mask-based energy function for scribble and point referring.

**Hard Mask-based Energy Function**    We first zero initialize a learnable latent variable $p_v$ with the same shape as $e_v$, and add it to the $e_v$. Then we can get $N$ attention maps from $N$ attention layers which model the relation between the context token and the novel visual tokens. Given the referring box or mask, we first convert it into a binary mask. Then, we compute the mask-based energy function based on the mask and the attention map $A^{(ct)}$, which is obtained by averaging pooling from $N$ attention maps. The energy function can be formulated as:

$$E\left(A^{(ct)}, r\right) = \left(1 - \frac{\sum_{i \in r} A_i^{(ct)}}{\sum_i A_i^{(ct)}}\right)^2,$$

(5)

where $r$ denotes the referring region. Then the gradient of the loss 5 is computed via backpropagation to update the learnable latent variable:

$$p_v \leftarrow p_v - \alpha \nabla_{p_v} E\left(A^{(ct)}, r\right),$$

(6)

where $\alpha > 0$ is a hyperparameter controlling the strength of the guidance. By optimizing $p_v$ through the Equation 6, we indirectly guide the attention maps to produce higher responses in the referring region $r$, thereby increasing the influence of the visual content of region $r$ on the output.

**Soft Mask-based Energy Function**    Since scribble and point lack the concept of the region, it is optional to use an extra SAM [31] model to obtain a mask for applying the Hard Mask-based Energy Function. However, this incurs additional inference cost, so we also provide an optional soft mask-based energy function based on a distance matrix $D$, which is computed via applying the OpenCV [7] *distanceTransform* function on the given scribble or point. Then the soft mask-based energy function can be formulated as:

$$E\left(A^{(ct)}, r\right) = \left(1 - \frac{\sum_{i \in r} \frac{e^{-D_i^2/2\sigma^2}}{\sqrt{2\pi}\sigma} A_i^{(ct)}}{\sum_i A_i^{(ct)}}\right)^2,$$

(7)

where $\sigma$ is the standard deviation of the Gaussian function, which is set to 0.1. By optimizing $p_v$ through the Equation 7, the closer the region of attention map is to the given scribble or point, the higher the response.

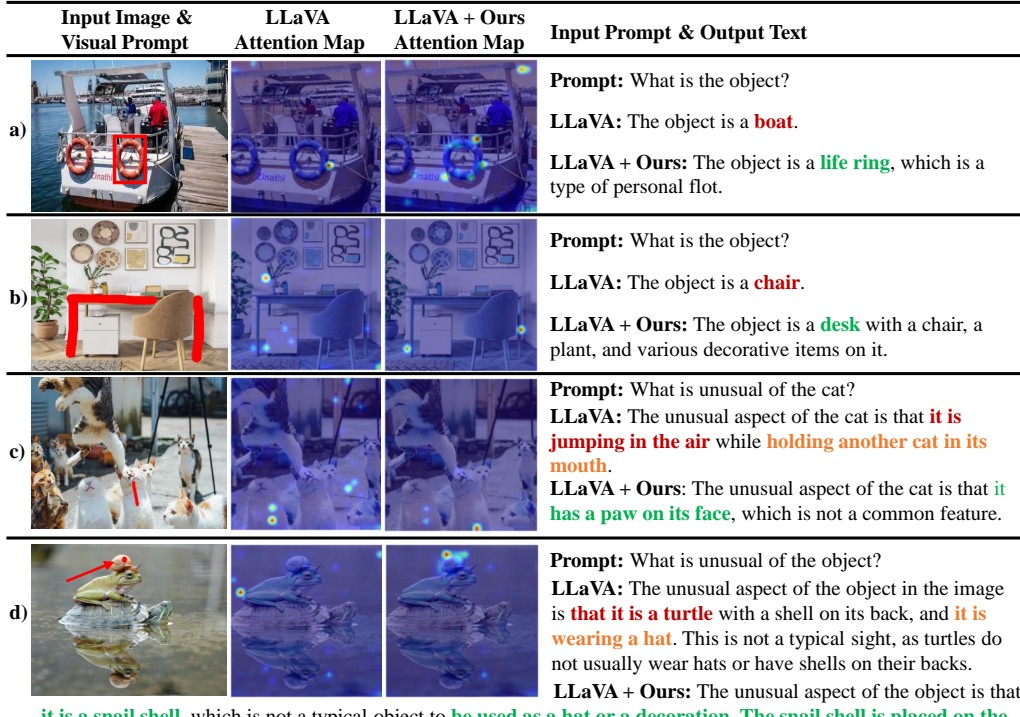

| | Input Image &
Visual Prompt | LLaVA
Attention Map | LLaVA + Ours
Attention Map | Input Prompt & Output Text |
|---|---|---|---|---|
| a) | | | | **Prompt:** What is the object?
**LLaVA:** The object is a **boat**.
**LLaVA + Ours:** The object is a **life ring**, which is a type of personal flot. |
| b) | | | | **Prompt:** What is the object?
**LLaVA:** The object is a **chair**.
**LLaVA + Ours:** The object is a **desk** with a chair, a plant, and various decorative items on it. |
| c) | | | | **Prompt:** What is unusual of the cat?
**LLaVA:** The unusual aspect of the cat is that **it is jumping in the air** while **holding another cat in its mouth**.
**LLaVA + Ours:** The unusual aspect of the cat is that it has a paw on its face, which is not a common feature. |
| d) | | | | **Prompt:** What is unusual of the object?
**LLaVA:** The unusual aspect of the object in the image is **that it is a turtle** with a shell on its back, and **it is wearing a hat**. This is not a typical sight, as turtles do not usually wear hats or have shells on their backs.
**LLaVA + Ours:** The unusual aspect of the object is that |

it is a snail shell, which is not a typical object to be used as a hat or a decoration. The snail shell is placed on the head of a frog, which is an unconventional combination of animals. This creates an interesting and unexpected scene, as it is not common to see a snail shell being used in such a manner.

Figure 5: The examples of referring MLLM with four types of visual prompt, including box (a), mask (b), scribble (c) and point (d). The correct referring expressions are marked in green, incorrect referring expressions are marked in red, and hallucinated expressions are marked in orange. Compared to the baseline model, our method enhances **interpretability** and **controllability** with visual prompts, while also helping the model **mitigate hallucination** issues.

Finally, we iteratively optimize the learnable latent variable $T$ times at the $0$-th step of model inference. In addition, to prevent overfitting, we employ Early Stop (ES) and Exponential Moving Average (EMA) strategies to enhance model stability. More details are shown in Appendix B.1.

# 5   Experiments

## 5.1   Experiment Details

Unless explicitly stated otherwise, the MLLM we use is LLaVA-v1.5-7B [35], $T$=5, $\alpha$=400 and $\beta = 0.5$. All experiments are conducted on two RTX 3090 GPUs with 24 GB of memory each.

## 5.2   Applications

**Referring with Different Visual Prompts.**    We first demonstrate referring QA with different visual prompts, including box, mask, scribble and point in the Figure 5. Our method consistently demonstrates significant controllability with four types of visual prompts. And our method improves the interpretability compared to basic model (column 3 vs column 2), demonstrates a stronger correlation between the attention response areas and the generated descriptions.

**Out-of-Domain Task.**    We present examples of the performance on out-of-domain tasks OCR and Screenshots. As shown in Figure 6, compared to Ferret, our method correctly identified the text in the referring region. Additionally, as shown in Figure 9, our method correctly recognized the app in the mobile screenshot, unlike Ferret.

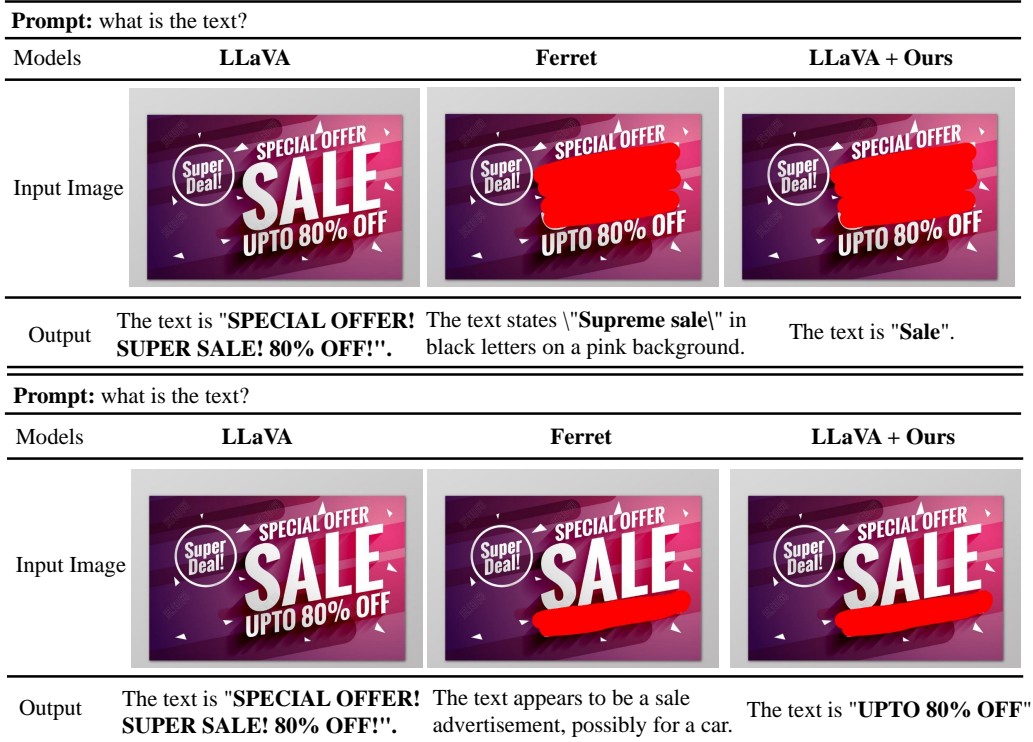

| **Prompt:** what is the text? | | |
| Models | **LLaVA** | **Ferret** | **LLaVA + Ours** |
| Input Image | | | |
| Output | The text is "**SPECIAL OFFER! SUPER SALE! 80% OFF!**". | The text states \"**Supreme sale**\" in black letters on a pink background. | The text is "**Sale**". |

| **Prompt:** what is the text? | | |
| Models | **LLaVA** | **Ferret** | **LLaVA + Ours** |
| Input Image | | | |
| Output | The text is "**SPECIAL OFFER! SUPER SALE! 80% OFF!**". | The text appears to be a sale advertisement, possibly for a car. | The text is "**UPTO 80% OFF**". |

Figure 6: Examples of comparing with training method Ferret on OCR.

**Impact on Hallucinations.** Our method guides the model to focus on specific regions, potentially helps the model mitigate hallucination issues, as shown in Figure 5 (c,d output in orange color).

### 5.3 Comparisons

**Comparison on Referring Object Classification Task.** Following Ferret [68, 74], we use the Referring Object Classification (ROC) task to evaluate whether our method can accurately pinpoint and understand the semantic of the referring region. The task requires the model to correctly identify the target within the referring region. We follow the setting of Ferret to form 1,748 questions (in which 1,548 for test and 200 for validation) based on LVIS [25] validation dataset, with corresponding box, mask, scribble and point. We consider the edit attention with $\eta = 10$ (as Equation 4 and Figure 3 (b)) as the baseline model. And we compare several training methods [43, 75, 12, 68]. We also evaluate the lower and upper limits of LLaVA's recognition capability by assessing LLaVA without referring region, as well as background blur outside the referring region, which are presented in gray. Additionally, we evaluate a method that highlights regions with color as a comparable training-free method. More details and the input examples are shown in Appendix B.2.

The results are shown in Table 1. Our method shows a better performance than the training method GPT4-ROI with box referring (60.59 vs 58.59) and the Shikra-7B with point referring (58.85 vs 56.27). However, due to the limitations of LLaVA's capabilities (as shown in the results of the LLaVA+Blur), we present a performance gap compared to the latest training method Ferret [68]. Our method also demonstrates superiority compared to training-free color prompt-based method and baseline method.

**Comparison on Referring Text Classification Task.** We consider the Referring Text Classification (RTC) task as the **out-of-domain** task, to verify the model's out-of-domain transfer capability. Similar to the ROC task, we formulate the problem as a binary classification task and construct 1,372 questions based on the COCO-Text [56] dataset. Since point and scribble referring methods are not

Table 1: The results on Referring Object Classification Task (test set). The prompt of the task is featured as "*Is the object ⟨location⟩ a ⟨class A⟩ or a ⟨class B⟩?*". "-" denotes the method does not support this type of referring. Results in gray font are provided for reference only.

| Models | Box | Mask | Scribble | Point |
|---|---|---|---|---|
| *Training Methods:* | | | | |
| Kosmos-2 [43] | 55.17 | - | - | - |
| GPT4-ROI [75] | 58.59 | - | - | - |
| Shikra-7B [12] | 64.60 | - | - | 56.27 |
| Ferret-7B [68] | 71.71 | 72.39 | 71.58 | 68.54 |
| *Training-Free Methods:* | | | | |
| LLaVA [36] | 54.72 | 54.72 | 54.72 | 54.72 |
| LLaVA + Blur | 73.39 | 71.32 | - | - |
| LLaVA + Color | 55.10 | 56.72 | - | - |
| LLaVA + Edit Att | 36.24 | 37.08 | - | - |
| LLaVA + **Ours** | **60.59** | **60.79** | **58.33** | **58.85** |

Table 2: The results on Referring Text Classification Task. The prompt of task is featured as "Is the text ⟨location⟩ of the image '⟨text A⟩' or '⟨text B⟩'?please select only one.".

| Models | Box | Mask |
|---|---|---|
| *Training Methods:* | | |
| Kosmos-2 [43] | 16.55 | - |
| GPT4-ROI [75] | 54.23 | - |
| Shikra-7B [12] | 50.07 | - |
| Ferret-7B [68] | 55.47 | 56.34 |
| *Training-Free Methods:* | | |
| LLaVA [36] | 53.57 | 55.47 |
| LLaVA + Blur | 83.60 | 74.49 |
| LLaVA + Color | 56.34 | 54.23 |
| LLaVA + Edit Att | 26.09 | 29.16 |
| LLaVA + **Ours** | **61.22** | **60.28** |

suitable for text due to the non-connectivity of the text, we only evaluate the RTC task with box and mask referring.

The results are shown in Table 2. All the training methods we evaluated exhibited poor out-of-domain generalization performance. Specifically, Ferret achieves only 55.47% accuracy on the RTC task, despite its excellent in-domain performance as shown in Table 1. In contrast, our training-free method still demonstrates the best **out-of-domain generalization** performance. We also present comparative examples of out-of-domain tasks, as shown in Figure 6 and Figure 9.

**More Tasks and MLLMs.** We also validate our method through the Referring Description Task on LLaVA-1.5-7B and InstructBLIP-7B [17]. The results are shown in Table 3. Our method consistently improves the model's referring description performance. And we validate our method on the more MLLMs through the ROC and RTC Tasks, MLLMs including InstructBLIP-7B and LLaVA-HR-7B [39], more details are shown in Appendix B.3. The results are shown in Table 4, our method consistently improves performance across different MLLMs. Due to InstructBLIP's relatively poor text recognition capabilities, our method results in only a modest improvement in the RTC task. However, thanks to the beneficial effect of image resolution on the RTC task, our method achieves a relative improvement of approximately 11.59% on LLaVA-HR.

### 5.4 Ablation Study

The ablation studies primarily focus on the box referred object classification. Furthermore, inspired by DIFNet [61], we calculate a relevancy between the model's output and pixels within the referring region to assess the extent to which the model's output is influenced by visual content within the region. More details and additional experiments are shown in Appendix B.2 and B.3.

**Impact of $T$ and $\alpha$.** As shown in Table 5, as $T$ increases, the relevancy between the model's output and the referring regions also increases. However, the larger $T$ results in a decrease in the model's accuracy on the ROC task, also with excessively large relevancy scores, showing that excessively large relevancy scores also indicate overfitting of the learnable latent variable. Therefore, the value of the relevancy score provides us with guidance to alleviate model overfitting, particularly when the relevancy score is around 0.18, typically resulting in better performance. And the value of $\alpha$ affects the convergence speed of optimization, with larger $\alpha$ also leading to overfitting of the model.

**Impact of EMA and ES.** As shown in Table 5, when equipped with a smaller $\beta$ value, it effectively mitigates the overfitting issue associated with the learnable latent variable. For instance, with $\alpha = 400$ and $\beta = 0.3$, the model's performance improves from 53.5 to 62.5. However, a smaller value of $\beta$ also results in slower convergence of the learnable latent variable. Therefore, we combine the Early Stop strategy, allowing us to use a slightly larger $\beta$ to accelerate the convergence of the learnable latent variable. After incorporating the early stop strategy, we can opt for slightly larger $T$ to ensure

Table 3: The results on box Referring Description Task on RefCOCOg [29]. The prompt of task is featured as "Can you provide a description of the region ⟨location⟩ in a sentence?".

| Models | B@4 | M | C | S |
|---|---|---|---|---|
| LLaVA [36] | 5.02 | 13.15 | 55.61 | 17.61 |
| LLaVA + Color | 5.37 | 11.57 | 55.27 | 17.01 |
| LLaVA + **Ours** | **5.53** | **14.00** | **59.75** | **19.08** |
| InstructBLIP [17] | 1.24 | 8.70 | 9.89 | 7.95 |
| InstructBLIP + Color | 1.27 | 8.26 | **14.16** | 6.92 |
| InstructBLIP + **Ours** | **1.39** | **8.77** | 10.28 | **8.24** |

Table 4: The results of combining with different MLLMs on ROC and RTC tasks (box, test set).

| Models | ROC | RTC |
|---|---|---|
| LLaVA [36] | 54.72 | 53.57 |
| LLaVA + Ours | **60.59** | **61.22** |
| InstructBLIP [17] | 49.81 | 26.46 |
| InstructBLIP + Ours | **54.91** | **28.94** |
| LLaVA-HR [39] | 53.81 | 47.01 |
| LLaVA-HR + Ours | **58.92** | **58.60** |

that the model is adequately optimized on challenging samples. The early stop strategy allows us to attain superior model performance while reducing the impact of overfitting. The additional experiment about ES is shown in Table 6.

**Impact of Different Text Prompts and the Size of Visual Prompts.** We explore the impact of different text prompts and the size of visual prompts in Figure 10 and Figure 11 respectively. The results demonstrate that combining a clear and specific text prompt with an appropriate visual prompt size typically leads to improved controllability.

# 6 Limitations

While we have demonstrated visual prompt control by optimizing only visual tokens, our technique is subject to a few limitations. First, there is some additional inference overhead, while various engineering approaches (such as Ollama [2]) can significantly speed up the process. Therefore, this limitation can be reasonably overlooked. Second, our method is applicable only to white-box models and relies on the basic capabilities of the models themselves. However, our approach is orthogonal to these ongoing advancements in foundational models. Third, currently, our method supports only a single region visual prompt, extending this to multi-region control is a direction for future work. Fourth, our current optimization strategy is relatively simple, and the selection of text prompts can also affect the optimization results. We plan to focus on improving this aspect in future research.

# 7 Conclusion

In this work, we present a training-free method to integrate visual prompts into Multimodal Large Language Models (MLLMs) through learnable latent variable optimization. By adjusting visual tokens during inference, our approach enhances the attention to referring regions, enabling detailed descriptions and reasoning without additional training costs. Our method supports various referring formats such as box, mask, scribble, and point. The results show that our approach demonstrates strong out-of-domain generalization and interpretability, making it a promising direction for embedding referring abilities into MLLMs.

## Acknowledgments and Disclosure of Funding

This work was supported by National Key R&D Program of China (No.2023YFB4502804), the National Science Fund for Distinguished Young Scholars (No.62025603), the National Natural Science Foundation of China (No. U22B2051, No. U21B2037, No. 62072389, No. 62302411), the Natural Science Foundation of Fujian Province of China (No.2021J06003), and China Postdoctoral Science Foundation (No. 2023M732948).

---

[2]https://ollama.com/

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

# A  Broader Impact

The integration of Multimodal Large Language Models (MLLMs) has far-reaching implications across various sectors. These models enhance accessibility, improve education, advance healthcare, and revolutionize media and entertainment. They offer intuitive interfaces for diverse communication needs, assist in medical diagnosis and treatment, and enable immersive multimedia experiences. However, ethical considerations must be addressed to ensure equitable and responsible deployment. Collaboration among researchers, policymakers, and industry is essential to maximize the societal impact of MLLMs.

# B  Appendix / supplemental material

## B.1  Details of EMA and ES

**EMA.**  Model weight Exponential Moving Average (EMA) is a technique used to stabilize the training of deep neural networks by maintaining a smoothed version of the model parameters. It calculates the moving average of the model weights by giving more weight to recent updates while gradually decreasing the influence of past updates. Mathematically, EMA is defined as:

$$\theta_{\text{EMA}}^{(t)} = \beta \cdot \theta^{(t)} + (1 - \beta) \cdot \theta_{\text{EMA}}^{(t-1)},$$

where $\theta_{\text{EMA}}^{(t)}$ is the EMA of the model weights at time $t$, $\theta^{(t)}$ is the model weights at time $t$, $\theta_{\text{EMA}}^{(t-1)}$ is the EMA of the model weights at the previous time step, $\beta$ is the smoothing factor, ranging from 0 to 1. EMA helps to stabilize the training process by reducing the variance of the parameter updates, which can prevent the model from diverging or oscillating during training. We employ the EMA on the learnable latent variable in our experiments during visual token optimization.

**ES.**  Early Stop (ES) is a regularization technique commonly used during the training of machine learning models to prevent overfitting. The main idea behind ES is to monitor the performance of the model on a validation set during training and stop the training process if the performance begins to deteriorate. Specifically, in our experiments, ES tracks the loss on the test sample and halts the visual token optimization process if the metric does not improve for a certain number of consecutive epochs. Mathematically, ES can be described as follows:

Let $L^{(i)}$ denote the loss at optimization process $t$, and let $\Delta$ represent a predefined threshold for acceptable loss degradation, then we

$$\text{Stop optimizing if } abs(L^{(t)} - L^{(0)})/L^{(0)} \geq \Delta \quad \text{for } T \text{ consecutive process.}$$

We set $\Delta = 0.25$ in our experiments. ES helps prevent overfitting by stopping the optimization process before the model starts to lose generalization ability. It provides a simple yet effective way to regularize the optimization process and improve the generalization performance of the model.

## B.2  Additional Experiment Details

**Details of Relevancy Score.**  To elucidate the influence of input visual pixels on outputs, we employ a technique known as the Relevancy Map. This map highlights the regions or features of the input data that contribute most significantly to the model's decision-making process. Specifically, the Relevancy Map assigns importance scores to different parts of the input, indicating their relative impact on the model's output. This interpretability tool not only enhances our understanding of the model's behavior but also facilitates error analysis and model debugging. More details can be found in the paper [61, 3, 9, 10, 50].

In our experiments, we provide the relevancy scores alongside model predictions to provide insights into the model's decision-making process. In MLLMs, the typical use of Key-Value cache technique in LLMs, for convenience, we propagate the relevance from the model's first output token to the input of LLM ( $e_v$ and $e_t$ ). Then the relevancy scores corresponding to visual tokens are reshaped into a grid that matches the layout of the original image. Since relevancy maps are akin to attention maps and often exhibit significantly higher values in localized regions, taking the average may dilute their

Table 5: The ablation of $T$, $\alpha$, EMA. We report Accuracy and Relevancy on ROC task (validation set). The best performance is highlighted in bold, while the paired Relevancy scores are indicated with underlines.

| $T \rightarrow$ | Accuracy | | | | | Relevancy | | | | |
|---|---|---|---|---|---|---|---|---|---|---|
| | 0 | 1 | 2 | 3 | 4 | 0 | 1 | 2 | 3 | 4 |
| $\alpha = 200$ | 57.00 | 57.50 | 59.00 | 60.50 | 58.00 | 0.1667 | 0.1679 | 0.1698 | 0.1743 | 0.2058 |
| $\alpha = 300$ | 57.00 | 57.50 | 60.00 | 59.00 | 58.50 | 0.1667 | 0.1684 | 0.1722 | 0.1986 | 0.2792 |
| $\alpha = 400$ | 57.00 | 58.50 | 60.50 | **61.00** | 53.50 | 0.1667 | 0.1689 | 0.1749 | 0.2238 | 2.5542 |
| $\alpha = 500$ | 57.00 | 59.50 | 60.50 | 56.00 | 43.00 | 0.1667 | 0.1694 | 0.1799 | 0.2650 | 0.4542 |
| w/ EMA ($\alpha = 400$) | | | | | | | | | | |
| $\beta = 0.3$ | 57.00 | 57.00 | 58.00 | 60.50 | **62.00** | 0.1667 | 0.1674 | 0.1689 | 0.1712 | 0.1753 |
| $\beta = 0.5$ | 57.00 | 57.50 | 60.00 | 61.00 | 60.00 | 0.1667 | 0.1679 | 0.1704 | 0.1767 | 0.2071 |
| $\beta = 0.7$ | 57.00 | 57.50 | 61.00 | **62.00** | 54.00 | 0.1667 | 0.1683 | 0.1728 | 0.1957 | 0.2992 |

Table 6: The ablation of ES ($\alpha = 400, \beta = 0.5$, validation set).

| $T \rightarrow$ | 0 | 4 | 5 |
|---|---|---|---|
| Acc. | 57.00 | 60.50 | 62.50 |
| Rel. | 0.1667 | 0.1841 | 0.1937 |

Table 7: Impact of LLM Size in Different MLLMs on ROC task (box, test set).

| Models | Vanilla | Ours |
|---|---|---|
| LLaVA-1.5-7B | 54.72 | **60.59** |
| LLaVA-1.5-13B | 55.69 | **58.40** |
| InstructBLIP-7B | 49.81 | **54.91** |
| InstructBLIP-13B | 54.33 | **59.24** |

significance. So we extract the max value in the referring region of relevancy map as final relevancy score.

It is also worth noting that we found relevancy map plays a similar role to attention map, directly modeling the relationship between input and output of the model. This suggests that we may be able to directly control the model's output based on relevancy map. However, calculating the relevancy map requires computing the gradient for each relevant tensor in the model. For convenience, the approach presented in this paper only utilizes attention for implementation, while the relevancy map only be used to assess the extent of visual impact on the output.

**Details of Implement.** We apply low-bit quantization to the basic model we implemented to further optimize memory usage. Nevertheless, we still achieve performance competitive with training-based methods (without quantization).

**Details and Input Examples of ROC Task.** The box and mask are from the LVIS GT boxes and mask, the scribble and the point are randomly sampled inside the boxes. **It is worth noting that we follow Ferret to choose negative object whose central point is close to the GT object. Although this is somewhat disadvantageous for us, we still achieve competitive performance compared to other methods.** And we show the input examples of different methods on ROC task in Figure 7.

### B.3 Additional Experiments

**Application on scene text recognition (OCR).** Results are shown in Figure 8. Our method can also perform referring region text recognition.

**Examples on Out-of-Domain Task.** We present examples of the performance on out-of-domain tasks OCR and Screenshots. As shown in Figure 6, compared to Ferret, our method correctly identified the text in the referring region. Additionally, as shown in Figure 9, our method correctly recognized the app in the mobile screenshot, unlike Ferret.

**Effect on More MLLMs.** We validate our method on various MLLMs, including LLaVA-1.5-7B, InstructBLIP-7B, and LLaVA-HR-7B. InstructBLIP employs a cross-attention image-text alignment module called Q-Former. Specifically, in InstructBLIP, the interaction between visual tokens and text tokens occurs within Q-Former, so we utilize the cross-attention map from Q-Former to optimize

Table 8: The inference cost with different actual output token numbers on a single GTX3090 GPU. LLaVA + Ours with $T = 5$ without using Early Stop here.

| Models | speed(s) | Max GPU Mem. |
|---|---|---|
| LLaVA (6 tokens) | 1.22 | 7G |
| LLaVA + Ours (7 tokens) | 3.56 | 14G |
| LLaVA (436 tokens) | 5.78 | 7G |
| LLaVA + Ours (439 tokens) | 7.45 | 14G |

visual tokens. LLaVA-HR supports input images with larger resolutions. The results are shown in Table 4.

**Comparison on Referring Description Task.**    We also validate our method through the Referring Description Task. Specifically, we construct the test set based on region-text pairs from the Ref-COCOg test split and evaluate the method using traditional captioning metrics BLEU@4 (B@4) [42], METEOR (M) [6], CIDEr-D (C) [55], and SPICE (S) [2]. However, it is important to note that these metrics are significantly influenced by the style and distribution of the model's output text. Typically, outputs that are more similar in distribution to RefCOCOg yield better results, while those with some unique styles lead to poorer results. Therefore, this experiment is only used for internal validation of the model and not for comparison between different models. The results are shown in Table 3. Our method consistently improves the model's referring description performance.

**Impact of LLM Size on Different MLLMs.**    We validate the impact of LLM size on LLaVA-1.5 and InstructBLIP, focusing on the 7B and 13B models. The results are shown in Table 7. Our method exhibited poorer performance in LLaVA-1.5-13B, which may be due to the increased number of attention maps, making the optimization of visual tokens more challenging. Therefore, it may be essential to adopt different hyperparameters for different models. In contrast, in InstructBLIP-7B and InstructBLIP-13B, our method consistently yielded performance improvements. This is likely because the interaction between visual tokens and text tokens occurs in Q-Former for InstructBLIP, thereby mitigating the optimization challenges associated with larger LLMs.

**Inference Cost.**    We compare the inference cost of our method and the basic LLaVA model. LLaVA + Ours model with $T = 5$ and does not use Early Stop here. Results are shown in Table 8, when outputting only 7 tokens, our method noticeably adds approximately 2 seconds of inference time. However, when generating about 400 tokens, the proportion of the extra inference time produced by our method significantly decreases. When combined with an early stop strategy, the proportion of additional inference time will be further reduced.

**Impact of Different Text Prompt.**    Results are shown in Figure 10. Different text prompts can significantly affect the performance of our method. For instance, our method fails when using an ambiguous text prompt like "describe the region in the image." However, it succeeds with a more specific referring text prompt such as "what is unusual about the region of the building?" Therefore, it is recommended to use clear and specific text prompts to achieve better control and performance.

**Impact of the Size of Visual Prompt.**    Results are shown in Figure 11. For box and mask prompts, when they do not fully cover the referring object, failure control may occur. This is because the highest attention response for the desired outcome may fall on any unexpected area of the object. Therefore, it is recommended to cover the object with a larger-sized visual prompt.

**Comparing Highlight Token and Context Token based Optimization.**    We compare Highlight Token and Context Token based Optimization as shown in Figure 12. Directly optimizing based on the highlight token is faster but also prone to overfitting. This may be due to the direct connection between the highlight token and visual token, while other text tokens contain redundant visual associations. However, this also ignores the potential influence of other text tokens.

**More Examples of Referring MLLMs with Scribble and Point.**    In Figure 14, we also show more examples of referring MLLMs with scribble (right) and point (left).

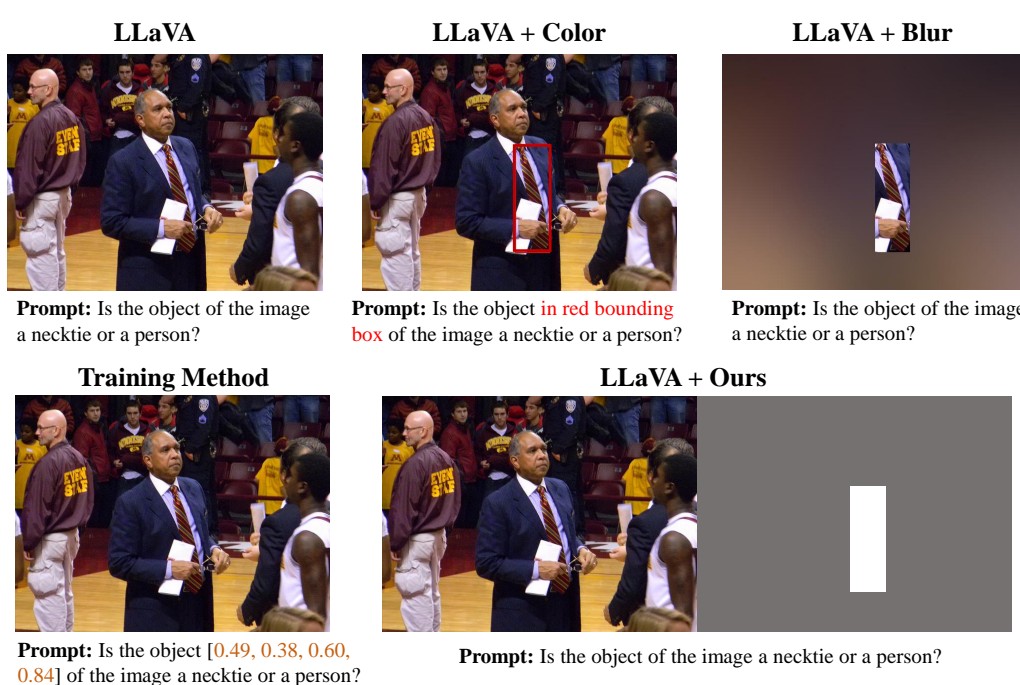

Figure 7: The input examples of ROC Task.

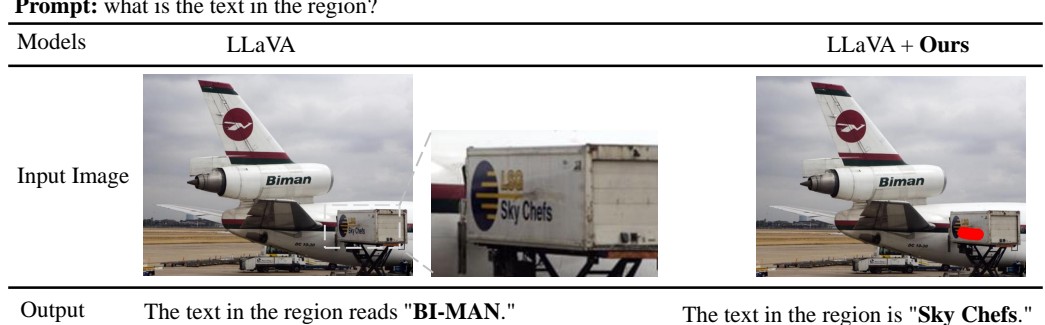

Figure 8: Application on natural scene text recognition (OCR).

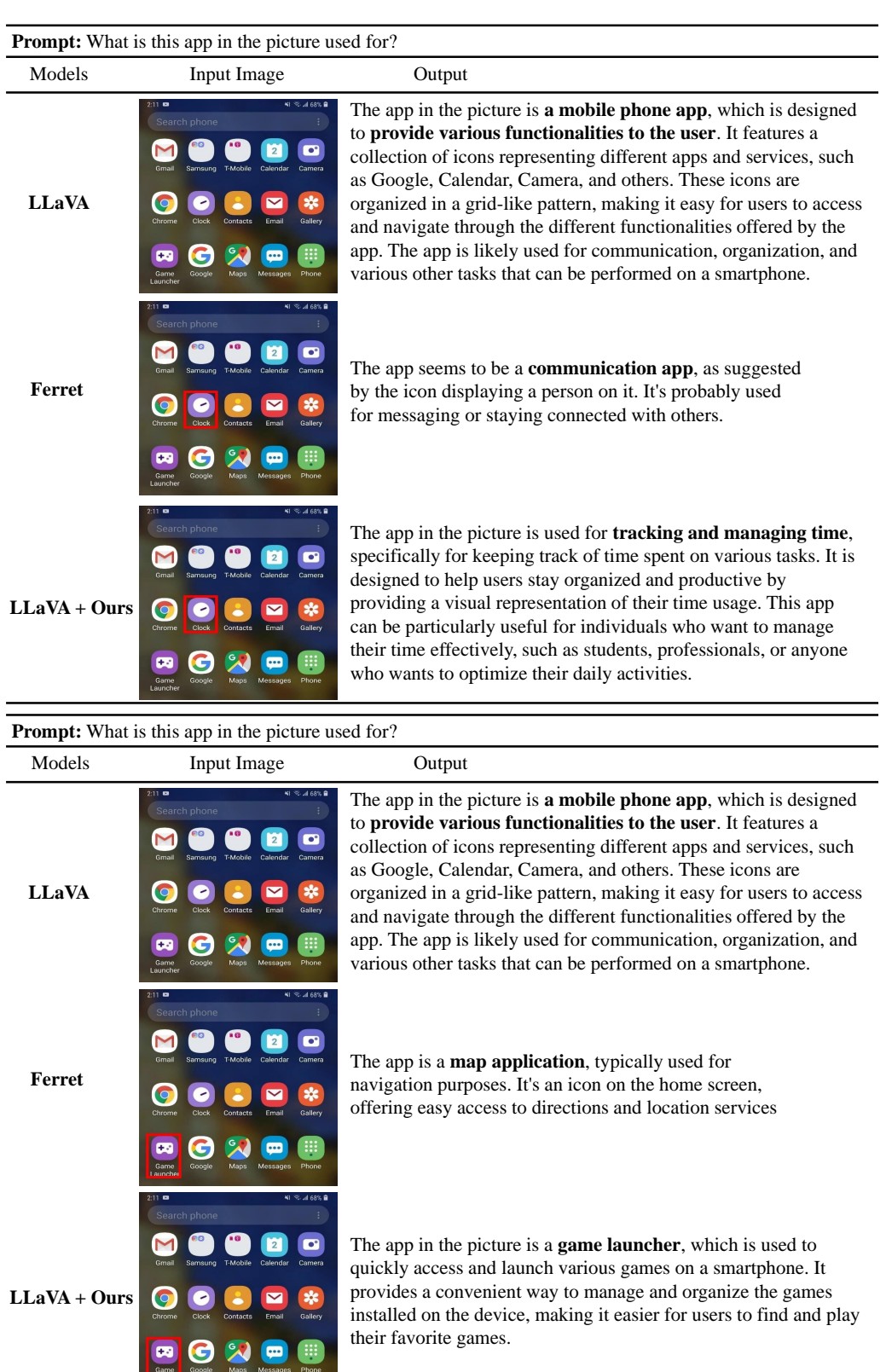

| **Prompt:** What is this app in the picture used for? | | |
|---|---|---|
| Models | Input Image | Output |
| LLaVA | | The app in the picture is **a mobile phone app**, which is designed to **provide various functionalities to the user**. It features a collection of icons representing different apps and services, such as Google, Calendar, Camera, and others. These icons are organized in a grid-like pattern, making it easy for users to access and navigate through the different functionalities offered by the app. The app is likely used for communication, organization, and various other tasks that can be performed on a smartphone. |
| Ferret | | The app seems to be a **communication app**, as suggested by the icon displaying a person on it. It's probably used for messaging or staying connected with others. |
| LLaVA + Ours | | The app in the picture is used for **tracking and managing time**, specifically for keeping track of time spent on various tasks. It is designed to help users stay organized and productive by providing a visual representation of their time usage. This app can be particularly useful for individuals who want to manage their time effectively, such as students, professionals, or anyone who wants to optimize their daily activities. |

| **Prompt:** What is this app in the picture used for? | | |
|---|---|---|
| Models | Input Image | Output |
| LLaVA | | The app in the picture is **a mobile phone app**, which is designed to **provide various functionalities to the user**. It features a collection of icons representing different apps and services, such as Google, Calendar, Camera, and others. These icons are organized in a grid-like pattern, making it easy for users to access and navigate through the different functionalities offered by the app. The app is likely used for communication, organization, and various other tasks that can be performed on a smartphone. |
| Ferret | | The app is a **map application**, typically used for navigation purposes. It's an icon on the home screen, offering easy access to directions and location services |
| LLaVA + Ours | | The app in the picture is a **game launcher**, which is used to quickly access and launch various games on a smartphone. It provides a convenient way to manage and organize the games installed on the device, making it easier for users to find and play their favorite games. |

Figure 9: Examples of comparing with training method Ferret on Screenshot.

**Prompt**: describe the region in the image.

| LLaVA |  | The image **features a group of people gathered outside a building**, possibly a church. There are four people in the scene, with two of them wearing ties, indicating a formal or semi-formal occasion. The group consists of a bride, a groom, and two other people. They are all standing together, possibly posing for a picture. The building has a black and white color scheme, adding a touch of elegance to the scene. |
|---|---|---|
| LLaVA + **Ours** |  | The image **features a group of people gathered outside a building**, possibly a church or a wedding venue. There are four people in the scene, with two of them wearing ties, suggesting a formal occasion. The group is standing on a brick walkway, and there are potted plants nearby, adding to the ambiance of the location. The people are engaged in conversation, creating a lively atmosphere. |

**Prompt**: what is unusual of the region of the building?

| LLaVA |  | The unusual aspect of the building's region is the **presence of a potted plant on the side of the building**. It is not common to see a potted plant on the side of a building, especially in a setting where people are dressed in formal attire, such as a wedding. The potted plant adds a unique touch to the scene and stands out among the other elements. |
|---|---|---|
| LLaVA + **Ours** |  | The unusual aspect of the building's region is the **presence of a large window in the middle of the structure**. This is not a common feature in most buildings, as windows are typically placed on the sides or corners of a building. The large window in the middle of the building adds an interesting architectural element to the structure. |

Figure 10: Impact of different text prompt. The clear and specific text prompt attains a better performance.

**Prompt:** what is unusual of the region of the building?

| | | |
|---|---|---|
| LLaVA | | The unusual aspect of the building's region is **the presence of a potted plant on the side of the building**. It is not common to see a potted plant on the side of a building, especially in a setting where people are dressed in formal attire, such as a wedding. The potted plant adds a unique touch to the scene and stands out among the other elements. |
| LLaVA + **Ours** | | The unusual aspect of the region of the building is **the presence of a potted plant on the ground**. Typically, potted plants are placed on elevated surfaces, such as tables or shelves, to provide better air circulation and prevent overwatering. However, in this image, the potted plant is located on the ground, which is not a common practice. This might be due to the specific needs of the plant or the design choice of the building's interior. |
| | | The unusual aspect of the building's region is **the presence of a large window in the middle of the structure**. This is not a common feature in most buildings, as windows are typically placed on the sides or corners of a building. The large window in the middle of the building adds an interesting architectural element to the structure. |
| | | The unusual feature of the region of the building is **the presence of a stained glass window**. Stained glass windows are not commonly found in modern buildings, especially in urban settings. This adds a unique and artistic touch to the building, making it stand out from other structures in the area. |

Figure 11: Impact of the size of visual prompt. The larger prompt size attains a better performance.

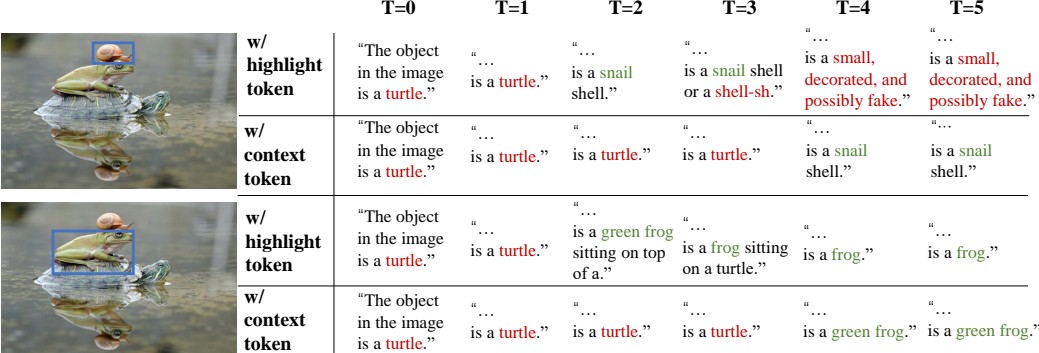

Prompt: "What's **object** in the image?"

Figure 12: Comparing highlight text token and context text token based optimization.

| | | T=0 | T=1 | T=3 | T=4 | T=5 |
|---|---|---|---|---|---|---|
| 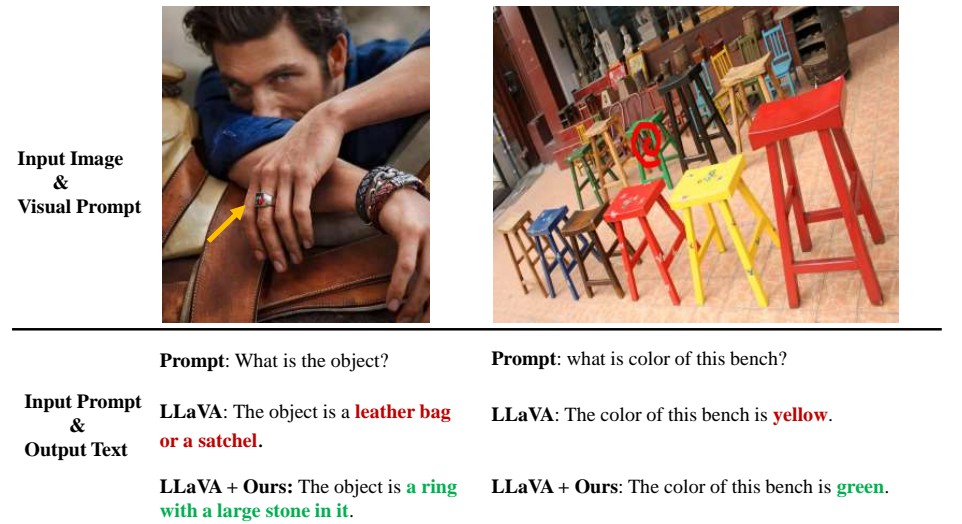 | w/o EMA | "The person wearing the hat is wearing a green hat." | "… a red hat." | "… a red hat." | "The color of the hat is red. The hues of the two pieces of the image are the same, and the hues of the two." | "The image of the image is a The image of the image is a The image of the image is a." |
| | w/ EMA | "The person wearing the hat is wearing a green hat." | "… a red hat." | "… a red hat." | "… a red hat." | "The color of the hat the person is wearing is red." |

Prompt: "What's color of hat the person wearing?"

Figure 13: Impact of EMA. The EMA stabilizes the optimization process.

| | | |
|---|---|---|
| **Input Image & Visual Prompt** | | |
| **Input Prompt & Output Text** | **Prompt**: What is the object?

**LLaVA**: The object is a **leather bag or a satchel**.

**LLaVA + Ours:** The object is **a ring with a large stone in it**. | **Prompt**: what is color of this bench?

**LLaVA**: The color of this bench is **yellow**.

**LLaVA + Ours**: The color of this bench is **green**. |

Figure 14: More examples of referring MLLMs with scribble (right) and point (left).

