# OpenReview forum: "ControlMLLM: Training-Free Visual Prompt Learning for Multimodal Large Language Models"
_NeurIPS.cc/2024/Conference — NeurIPS 2024 poster_

### Official Review · Reviewer_LggR · 2024-06-30

**Soundness:** 2
**Presentation:** 1
**Contribution:** 1
**Rating:** 3
**Confidence:** 5

**Summary:**

This paper proposes a method to integrate visual prompts into MLLMs without requiring additional training. The key idea is to optimize a learnable latent variable to enhance the attention response to visual tokens during inference, thereby improving the model's ability to focus on specific regions in the visual input.

**Strengths:**

The proposed approach does not requires additional training for unseen datasets.

**Weaknesses:**

1. Very poor writing. The use of abbreviations and full terms is very inconsistent. For example, "Multimodal Large Language Models" is spelled out in full on Line 29-30, while the abbreviation "MLLMs" is used frequently earlier in the text. Additionally, "Linguistic" in Line 34 should be "textual." There are many grammatical errors in Figure 's prompt: "What's color of hat the person wearing?" and output: "The person wearing the hat is wearing a green hat.". Even the title in section 4.2 contains grammatical errors.
2. This paper lacks novelty and is an incremental form of previously proposed methods with no innovative points.
3. In Line 3, the authors claim that attention connects visual tokens and textual tokens, but in Line 33, it changes to MLP.
4. The experimental results are insufficient and lack numerous baselines, such as LLaVA1.5[1], LLaVA-NeXT[2], Monkey[3], and Qwen-VL[4].
5. The motivation for this study is insufficient. I think there is a baseline that by making the prompt descriptions clearer and more comprehensive based on the original MLLM. This baseline can also leverage MLLM's inherent ability to focus on specific regions.

[1] Improved Baselines with Visual Instruction Tuning.

[2] LLaVA-NeXT: Improved reasoning, OCR, and world knowledge.

[3] Monkey: Image Resolution and Text Label Are Important Things for Large Multi-modal Models.

[4] Qwen-VL: A Versatile Vision-Language Model for Understanding, Localization, Text Reading, and Beyond

**Questions:**

See weakness.

**Limitations:**

See weakness.

---

> ### Author Rebuttal · Authors · 2024-08-06
>
> # Rebuttal
>
> >#### **Q1: The use of abbreviations and full terms is very inconsistent. For example, "Multimodal Large Language Models" is spelled out in full on Line 29-30, while the abbreviation "MLLMs" is used frequently earlier in the text. Additionally, "Linguistic" in Line 34 should be "textual." There are many grammatical errors in Figure 's prompt: "What's color of hat the person wearing?" and output: "The person wearing the hat is wearing a green hat.". Even the title in section 4.2 contains grammatical errors.**
>
> We have provided the full term "Multimodal Large Language Models" and its abbreviation "MLLMs" in earlier text，such as lines 2 and 19. Thank you for pointing out the need for consistent terminology and other writing issues. We will correct these in the final version, including using "textual" instead of "linguistic" and addressing the grammatical errors.
>
> >#### **Q2: This paper lacks novelty and is an incremental form of previously proposed methods with no innovative points.**
>
> Thank you for your comments. We appreciate the opportunity to reiterate the innovations of our paper. Firstly, our motivation is novel. This paper introduces a setting for embedding visual prompts without additional training, enabling the integration of visual prompts into existing models conveniently. On the technical side, we leverage the idea of visual prompts and introduce a test-time prompt tuning strategy to adjust the attention distribution of MLLMs and facilitate the injection of visual prompts. This technique also offers valuable insights for improving the interpretability of MLLMs. Our contributions have been recognized by Reviewer VBnv as well.
>
> We will rephrase the contributions in the revised version, and your suggestions will help make our paper clearer. Thank you for your valuable feedback.
>
> >#### **Q3: In Line 3, the authors claim that attention connects visual tokens and textual tokens, but in Line 33, it changes to MLP.**
>
> In lines 5 and 34, we explain that specific MLP layers can influence the attention responses between visual and textual tokens. We provide a detailed analysis of this in Section 4.1, demonstrating how MLP outputs can control the interaction between these tokens.
>
> >#### **Q4: The experimental results are insufficient and lack numerous baselines, such as LLaVA1.5, LLaVA-NeXT, Monkey, and Qwen-VL.**
>
> Thank you for your constructive feedback. Our primary baseline is LLaVA1.5, as indicated in the "Experiment Details" section. We also present experiments on InstructBLIP in Table 5. Following your suggestion, we provide additional results in the following table. Qwen-VL is a trained referring MLLM, and LLaVA-Next is a recently released project supporting high-resolution input, both of which are classic and influential works. However, due to time and cost constraints, we conducted experiments on LLaVA-HR [1] as an alternative to LLaVA-Next. In the final version, we will include all these methods, including LLaVA-Next, for discussion and comparison
>
>  | Model   | Task | Vanilla | Ours  |
> |---------------|------|---------|-------|
> |Training method||||
> |**Qwen-VL**|ROC|72.6|-|
> | |RTC|64.7|-|
> |Training-free method||||
> | **LLaVA1.5**  | ROC  | 54.72   | 60.59 |
> |               | RTC  | 53.57   | 61.22 |
> | **InstructBLIP** | ROC  | 49.81   | 54.91 |
> |               | RTC  | 26.46   | 28.94 |
> | **LLaVA-HR**  | ROC  | 53.81   | 58.92 |
> |               | RTC  | 47.01   | 58.60 |
> | **Monkey**    | ROC  | 55.26   | 60.68 |
> |               | RTC  | 55.59   | 63.39 |
>
> [1] Feast Your Eyes: Mixture-of-Resolution Adaptation for Multimodal Large Language Models.
>
> >#### **Q5: The motivation for this study is insufficient. I think there is a baseline that by making the prompt descriptions clearer and more comprehensive based on the original MLLM. This baseline can also leverage MLLM's inherent ability to focus on specific regions.**
>
> Our primary motivation is to introduce referring capabilities to MLLMs without significant training costs, rather than just proposing the task of injecting referring capabilities. Regarding the baseline suggested, it primarily serves to demonstrate the necessity of visual prompts. As discussed in Section 2.2, extensive prior work on training referring MLLMs has validated the importance of visual prompts. They simplify the referring process for users, reducing the need for precise text prompts and improving interaction efficiency. Moreover, clearly describing specific regions through text prompts can be challenging and unfriendly for users. Constructing such a baseline requires substantial effort, which is impractical.
>
> Thank you for your feedback. Indeed, the value of "referring MLLMs" has been explored and studied by many researchers [2-7], and this is not the primary contribution of our paper. The significance of this task lies in addressing scenarios that are challenging for language instructions alone, serving as a valuable complement to them. Constructing the baseline you mentioned is currently quite challenging and a key area for future research, which is one of the significant aspects of the referring MLLMs task.
>
> Our core contribution is in embedding visual prompt information into MLLMs without additional training, which is highly innovative for the field. This approach enables any MLLM to gain referential capabilities plug-and-play while maintaining generalizability. We hope this clarifies our contribution and addresses your concerns.
>
> [2] Qwen-vl: A versatile vision-language model for understanding, localization, text reading, and beyond.
>
> [3] Ferret: Refer and ground anything anywhere at any granularity
>
> [4]  Draw-and-understand: Leveraging visual prompts to enable mllms to comprehend what you want
>
> [5] Kosmos-2: Grounding multimodal large language models to the world
>
> [6] Shikra: Unleashing multimodal llm’s referential dialogue magic
>
> [7] Ferret-ui: Grounded mobile ui understanding with multimodal llms

---

> > ### Comment · Area_Chair_tEJ7 · 2024-08-12
> >
> > Hello Reviewer,
> >
> > The author has submitted a response to your comments. Whether or not it addresses your concerns, it would be greatly appreciated if you could acknowledge that you have reviewed the reply.

---

> > ### Comment · Reviewer_LggR · 2024-08-13
> >
> > Thanks to the authors' response. Although some of my concerns have been addressed, I believe there is still significant room for improvement and enhancement in this work. Therefore, I will maintain my score.

---

### Official Review · Reviewer_1iuY · 2024-07-10

**Soundness:** 3
**Presentation:** 3
**Contribution:** 2
**Rating:** 4
**Confidence:** 4

**Summary:**

The paper introduces a training-free approach to improve the referring capabilities of multimodal large language models (MLLM). In particular, the authors iteratively adjust the attention maps using a learnable latent variable, which is based on energy functions. They empirically validate the efficacy of their method on referring classification tasks, leveraging the foundation of LLaVA.

**Strengths:**

+ The paper is generally easy to read and follow.
+ The figures are clear, especially the visualization of attention maps under different conditions.
+ The proposed method is training-free and theoretically plug-to-play with different foundation models.

**Weaknesses:**

1. The generalization ability of the proposed method has not been fully verified. (a) It's uncertain whether the method can be applied to MLLMs beyond LLaVA. As the foundational model strengthens, the method's effectiveness could potentially diminish. (b) The energy function, based on a soft mask, heavily relies on the quality of segmentation models. (c) As seen in Table 3, the results appear to be sensitive to hyperparameters. This raises the question: do we need to meticulously adjust the parameters for each model and sample? If that's the case, the practicality of this method could be questionable.

2. There is a naive baseline that requires discussion: directly cropping the referred region and feeding it into the LLM.

3. In Tab. 2&3, the tasks and compared models are indeed limited. It is recommended to discuss more state-of-the-art MLLMs on wider tasks and datasets.

**Questions:**

Please see the Weaknesses.

**Limitations:**

The authors have discussed it in Sec. 6.

---

> ### Author Rebuttal · Authors · 2024-08-06
>
> # Rebuttal
> >#### **Q1: The generalization ability of the proposed method has not been fully verified. (a) It's uncertain whether the method can be applied to MLLMs beyond LLaVA. As the foundational model strengthens, the method's effectiveness could potentially diminish. (b) The energy function, based on a soft mask, heavily relies on the quality of segmentation models. (c) As seen in Table 3, the results appear to be sensitive to hyperparameters. This raises the question: do we need to meticulously adjust the parameters for each model and sample? If that's the case, the practicality of this method could be questionable.**
>
> (a) We have included more results, such as InstructBLIP model and Referring Description task in Tables 4 and 5 to demonstrate the generalization ability of our method. Our approach fundamentally enhances MLLMs, which primarily possess classification capabilities, by endowing them with referring abilities. Specifically, our method provides MLLMs with localization capabilities rather than relying solely on classification. Since classification relies on the foundational model, our method is orthogonal to the foundational model. As seen in Table 5, the less significant improvement on LLaVA-13b compared to LLaVA-7b is due to the increased complexity of the LLaVA-13b decoder, which makes optimization more challenging. Thus, the effectiveness of our method is related to the difficulty of optimizing the model, not necessarily diminished by the enhancement of the foundational model.
>
> Following your suggestions, we have included several methods beyond LLava, as detailed in Q3. We will create a dedicated section to discuss this area, which will make our paper's contributions even more robust.
>
> (b) Our method only requires an additional visual prompt during inference and does not depend on segmentation models, although they could be an optional technical route. In our approach, the soft mask is calculated using OpenCV functions, ensuring it is not heavily reliant on segmentation models' quality.
>
> (c) As demonstrated in Table 5, we conducted experiments on the InstructBLIP model using the same hyperparameters as LLaVA, achieving superior performance. Although fine-tuning the hyperparameters could lead to even better results, we plan to further improve our optimization strategy to make it more adaptable to different models.
>
> >#### **Q2: There is a naive baseline that requires discussion: directly cropping the referred region and feeding it into the LLM.**
>
> Thanks for your constructive suggestion. Following your suggestion, we have provided relevant results for reference in the following table. Given that LLaVA's image preprocessing resizes images to 224x224, the classification performance on small objects might be better, potentially resulting in superior performance compared to LLaVA + Blur. We will add the above discussion into the new version.
>
> | Model           | ROC   | RTC   |
> |-----------------|-------|-------|
> | **LLaVA**       | 54.72 | 53.57 |
> | **LLaVA + Blur**| 73.39 | 83.60 |
> | **LLaVA + Crop**| 82.04 | 88.78 |
> | **LLaVA + Ours**| 60.59 | 61.22 |
>
> >#### **Q3: In Tab. 2&3, the tasks and compared models are indeed limited. It is recommended to discuss more state-of-the-art MLLMs on wider tasks and datasets.**
>
> Thanks for your great commnet. We have included results for the RD task and the InstructBLIP model in Tables 4 and 5. It is also noteworthy that in the RD task, the performance differences among various foundational models are significantly influenced by the task and evaluation strategies. In contrast, in binary classification tasks like ROC and RTC, the results from different foundational models are relatively robust. This is why we followed Ferret in primarily experimenting on ROC and RTC tasks. We also provide more results in the following table.
>
>  | Model         | Task | Vanilla | Ours  |
> |---------------|------|---------|-------|
> | **LLaVA1.5**  | ROC  | 54.72   | 60.59 |
> |               | RTC  | 53.57   | 61.22 |
> | **InstructBLIP** | ROC  | 49.81   | 54.91 |
> |               | RTC  | 26.46   | 28.94 |
> | **LLaVA-HR**  | ROC  | 53.81   | 58.92 |
> |               | RTC  | 47.01   | 58.60 |
> | **Monkey**    | ROC  | 55.26   | 60.68 |
> |               | RTC  | 55.59   | 63.39 |

---

> > ### Comment · Area_Chair_tEJ7 · 2024-08-12
> >
> > Hello Reviewer,
> >
> > The author has submitted a response to your comments. Whether or not it addresses your concerns, it would be greatly appreciated if you could acknowledge that you have reviewed the reply.

---

> ### Author Response · Authors · 2024-08-13
>
> Dear Reviewer,
>
> Thank you for taking the time to review our paper and offer your valuable suggestions. We have thoroughly addressed your concerns in the rebuttal. We kindly ask if you would consider raising the score for our paper.

---

### Official Review · Reviewer_hD7U · 2024-07-11

**Soundness:** 3
**Presentation:** 2
**Contribution:** 3
**Rating:** 5
**Confidence:** 4

**Summary:**

This paper introduces a training-free approach to integrate visual prompts into MLLMs using learnable latent variables, aiming to enhance the model's interpretability and generalization. It adjusts visual tokens from MLP outputs and optimizes latent variables with an energy function to improve attention on relevant visual regions.

**Strengths:**

1.	This paper demonstrates and visualizes how the attention between prompt tokens and visual tokens differs across various layers.
2.	Figures 2 and 4 are helpful to understand the method.
3.	The quantitative and visualization experiments validate the effectiveness of the proposed approach.

**Weaknesses:**

1. Figures 3(a) and 3(b) show visualization results for different values of η. The paper mentions that in 3(a), η is too small to effectively control the attention. However, the focus of attention map in 3(b) does not significantly differ from that in 3(a).
2. Table 3 shows that the highest accuracy is achieved when α = 400 and T = 3. Why, then, is the value of T ultimately chosen as 4?
3. Since there are no recent developments, contributions, or updates to the MLLM, its detailed presentation in Eq.s (1)-(3) might be unnecessary and could be omitted. Besides, some symbols are not defined, such as, $I_i$, $A_i^{(ct)}$.

**Questions:**

1. It is recommended to validate the effectiveness of the method on additional MLLMs.
2. Additionally, please note that the title listed on the paper submission does not match the title in the PDF.

**Limitations:**

The limitations have been discussed.

---

> ### Author Rebuttal · Authors · 2024-08-06
>
> # Rebuttal
>
> >#### **Q1: Figures 3(a) and 3(b) show visualization results for different values of $\eta$. The paper mentions that in 3(a), $\eta$ is too small to effectively control the attention. However, the focus of attention map in 3(b) does not significantly differ from that in 3(a).**
>
>
> Thank you for your question. To clarify, there is a noticeable difference in the response of the green hat region between Figures 3(a) and 3(b). This indicates that in certain layers, the attention in the green hat region has a higher response, and these layers may play a crucial role in determining the model's output. It is important to note that not all layers' visual tokens determine the model's output[1]. Some layers might purely serve the purpose of organizing language, which explains why we cannot achieve our goal by editing the attention in all layers directly.
>
> However, our approach can implicitly guide the model to cause changes in the attention responses in certain layers without excessively affecting the model's ability to organize language. We will clarify this point in the final version.
>
> >#### **Q2: Table 3 shows that the highest accuracy is achieved when $\alpha$ = 400 and T = 3. Why, then, is the value of T ultimately chosen as 4?**
>
> As outlined in the 'Impact of EMA and ES' section, we combined EMA (Exponential Moving Average) and ES (Early Stopping) strategies to enhance the stability and convergence speed of model optimization. We observed in Table 6 that a larger $T$ value, when combined with ES, led to better results. Therefore, we chose a slightly larger $T = 4$ to ensure adequate optimization for more challenging samples, even if it involved a slight trade-off in peak accuracy. This decision was based on our observation that a $T$ value of 4 and a relevancy score around 0.18 helped maintain consistency and robustness across various validation scenarios.
>
> >#### **Q3: Since there are no recent developments, contributions, or updates to the MLLM, its detailed presentation in Eq.s (1)-(3) might be unnecessary and could be omitted. Besides, some symbols are not defined, such as, $I_i$ and $A_i^{(ct)}$.**
>
> We understand the importance of clarity and will refine the manuscript to ensure that all terms are well-defined. Additionally, we will reassess the inclusion of these equations to ensure that the presentation remains concise and focused on novel contributions.
>
> >#### **Q4: It is recommended to validate the effectiveness of the method on additional MLLMs.**
>
> As demonstrated in Table 5, we have already validated our method on several additional models, such as InstructBLIP. These experiments provide preliminary evidence of the method's generalizability. We also provide more results in the following table.
>
> | Model         | Task | Vanilla | Ours  |
> |---------------|------|---------|-------|
> | **LLaVA1.5**  | ROC  | 54.72   | 60.59 |
> |               | RTC  | 53.57   | 61.22 |
> | **InstructBLIP** | ROC  | 49.81   | 54.91 |
> |               | RTC  | 26.46   | 28.94 |
> | **LLaVA-HR**  | ROC  | 53.81   | 58.92 |
> |               | RTC  | 47.01   | 58.60 |
> | **Monkey**    | ROC  | 55.26   | 60.68 |
> |               | RTC  | 55.59   | 63.39 |
>
> >#### **Q5: Additionally, please note that the title listed on the paper submission does not match the title in the PDF.**
>
> Thank you for pointing this out. The discrepancy arose due to a last-minute change in the paper title after the abstract submission deadline. We will ensure that the final version of the paper has a consistent title across all submission materials to avoid confusion.
>
> [1] An Image is Worth 1/2 Tokens After Layer 2: Plug-and-Play Inference Acceleration for Large Vision-Language Models.

---

> > ### Comment · Area_Chair_tEJ7 · 2024-08-12
> >
> > Hello Reviewer,
> >
> > The author has submitted a response to your comments. Whether or not it addresses your concerns, it would be greatly appreciated if you could acknowledge that you have reviewed the reply.

---

> ### Author Response · Authors · 2024-08-13
>
> Dear Reviewer,
>
> Thank you for taking the time out of your busy schedule to review our paper and provide valuable feedback. We have addressed your concerns in the rebuttal. We kindly ask if you would consider raising the score for our paper.

---

### Official Review · Reviewer_VBnv · 2024-07-12

**Soundness:** 3
**Presentation:** 4
**Contribution:** 4
**Rating:** 6
**Confidence:** 4

**Summary:**

This paper proposes a training-free ControlMLLM, which uses optimizable latent variables to inject visual prompts into multimodal large MLLMs. The core idea is to adjust the visual token outputs of the MLP during inference, to control the attention response and ensure that the text prompt tokens focus on the indicated visual regions. It enhances the intensity of the indicated regions in the attention maps by optimizing learnable latent variables based on an energy function, enabling reference to various visual prompts (including boxes, masks, scribbles, and points) without model training, fine-tuning, or additional data. The method demonstrates out-of-domain generalization and interpretability, providing a promising direction for integrating referential capabilities into MLLMs. Experiments show that the proposed model is effective.

**Strengths:**

1), Prompt tuning for MLLMs is an interesting direction, and the proposed ideas are both effective and simple. Motivated by text-to-image works, ControlMLLM aims to control the attention map between the textual tokens and visual patches. This idea makes sense and provides a new direction to improve MLLMs.

2), The comparisons and ablations show the effectiveness of the proposed model.

3), The writing is clear, and the figures and tables help to understand the motivations.

**Weaknesses:**

1), The visual prompt shows great improvements over the base model LLaVA, however, it requires additional guidance information and more inference time, which may limit the applications of ControlMLLM. Especially in some complex scenarios where the guidance signals are unavailable.

2), In addition, the region signal $r$ plays a core role during optimization, and it controls the output of MLLMs. What if $r$ itself is wrong?  It may mislead the MLLMs.

**Questions:**

See above

---

> ### Author Rebuttal · Authors · 2024-08-06
>
> # **Rebuttal**
>
> >#### ****Q1: The visual prompt shows great improvements over the base model LLaVA, however, it requires additional guidance information and more inference time, which may limit the applications of ControlMLLM. Especially in some complex scenarios where the guidance signals are unavailable.****
>
> Thank you for your comment. Indeed, all settings that apply MLLMs to visual grounding, such as Referring MLLMs, require visual prompts [1, 2, 3, 4, 5, 6], this is not unique to our approach. However, our setup allows visual prompts to be injected into MLLMs without additional training, offering greater flexibility. Additionally, our method does not require fine-tuning the large model, which provides stronger generalizability. These features make our model better suited for transfer to other complex scenarios. As shown in Table 2, our method can easily transfer referential capabilities to out-of-domain data.
>
> Regarding the time issue, the comparison is 5.78s vs 7.45s when model output about 400 tokens, as shown in Table 7. Our model is indeed slower, but the difference is acceptable. We will further optimize this aspect in future work.
>
> >#### ****Q2:  In addition, the region signal plays a core role during optimization, and it controls the output of MLLMs. What if itself is wrong? It may mislead the MLLMs.****
>
> Thank you for your insightful comments and valuable suggestions. Your feedback has greatly guided our research direction. Visual prompts and text prompts, which are user-provided instructions during inference, play a crucial role in Human-Computer Interaction. While users should accurately specify the desired region, we acknowledge the risk of incorrect input potentially misleading the MLLMs. To address this, we propose incorporating validation mechanisms or fallback options in future versions to ensure robustness even when faced with inaccurate prompts. This approach aims to improve the reliability of the method in real-world applications.
>
> [1] Ferret: Refer and ground anything anywhere at any granularity
>
> [2]  Draw-and-understand: Leveraging visual prompts to enable mllms to comprehend what you want
>
> [3] Kosmos-2: Grounding multimodal large language models to the world
>
> [4] Shikra: Unleashing multimodal llm’s referential dialogue magic
>
> [5] Ferret-ui: Grounded mobile ui understanding with multimodal llms
>
> [6] Qwen-vl: A versatile vision-language model for understanding, localization, text reading, and beyond.

---

> > ### Comment · Reviewer_VBnv · 2024-08-12
> >
> > Thanks to the authors for their response, and I read the other comments as well. This paper makes an interesting idea of training-free prompt tuning for MLLM. As a result, I have decided to raise my rating.

---

> > > ### Author Response · Authors · 2024-08-13
> > >
> > > Dear Reviewer,
> > >
> > > Thank you for carefully reading our paper and rebuttal, and for recognizing our work. We wish you the best of luck.

---

### Decision · Program_Chairs · 2024-09-25

**Decision:**

Accept (poster)

**Comment:**

This paper proposes a training-free Control MLLM. The core idea is to adjust the visual token outputs of the MLP during inference to control the attention, ensuring that the text prompt tokens focus on the specified visual regions. The model enhances the intensity of the indicated regions in the attention maps by optimizing learnable latent variables based on an energy function. This enables the model to refer to various visual prompts (including boxes, masks, scribbles, and points) without requiring fine-tuning or additional data. Experiments demonstrate that the proposed model is effective. A reviewer questioned the generalization capability of the proposed approach, to which the authors responded in their rebuttal.